# Differences between blacks and whites in well-being, beliefs, emotional states, behaviors and survival, 1978-2014

Zafar Zafari[1]*, Katherine M. Keyes[2], Boshen Jiao[3], Sharifa Z. Williams[4], Peter Alexander Muennig[5]

**1** Department of Pharmaceutical Health Services Research, University of Maryland School of Pharmacy, Baltimore, Maryland, United States of America, **2** Department of Epidemiology, Mailman School of Public Health, Columbia University, New York, New York, United States of America, **3** The Comparative Health Outcomes, Policy, and Economics (CHOICE) Institute, University of Washington, Seattle, Washington, United States of America, **4** Center for Research on Cultural and Structural Equity in Behavioral Health, Nathan S. Kline Institute for Psychiatric Research, Orangeburg, New York, United States of America, **5** Global Research Analytics for Population Health, Health Policy and Management, Columbia University, New York, New York, United States of America

* zzafari@rx.umaryland.edu

**Data Availability Statement:** The dataset we used, the General Social Survey-National Death Index, was determined to contain sensitive and potentially identifiable information by its host. The National

## Abstract

### Objectives

Material well-being, beliefs, and emotional states are believed to influence one's health and longevity. In this paper, we explore racial differences in self-rated health, happiness, trust in others, feeling that society is fair, believing in God, frequency of sexual intercourse, educational attainment, and percent in poverty and their association with mortality.

### Study designs

Age-period-cohort (APC) study.

### Methods

Using data from the 1978–2014 General Social Survey-National Death Index (GSS-NDI), we conducted APC analyses using generalized linear models to quantify the temporal trends of racial differences in our selected measures of well-being, beliefs, and emotional states. We then conducted APC survival analysis using mixed-effects Cox proportional hazard models to quantify the temporal trends of racial differences in survival after removing the effects of racial differences in our selected measures.

### Results

For whites, the decline in happiness was steeper than for blacks despite an increase in high school graduation rates among whites relative to blacks over the entire period, 1978–2010. Self-rated health increased in whites relative to blacks from 1978 through 1989 but underwent a relative decline thereafter. After adjusting for age, sex, period effects, and birth cohort effects, whites, overall, had higher rates of self-rated health (odds ratio [OR] = 1.88;

Opinion Research Center (NORC) has determined that they are identifiable data and has placed restrictions upon their use. The access instructions can be viewed at: https://gss.norc.org/Documents/other/ObtainingGSSSensitiveDataFiles.pdf.

**Funding:** This study was funded by Global Research Analytics for Population Health at the Mailman School of Health, Columbia University. The funders had no role in study design, data collection and analysis, decision to publish, or preparation of the manuscript.

**Competing interests:** The authors have declared that no competing interests exist.

95% confidence interval [CI] = 1.63, 2.16), happiness (OR = 2.05; 1.77, 2.36), and high school graduation (OR = 2.88; 2.34, 3.53) compared with blacks. Self-rated health, happiness, and high school graduation also mediated racial differences in survival over time.

## Conclusions

We showed that some racial differences in survival could be partly mitigated by eliminating racial differences in health, happiness, and educational attainment. Future research is needed to analyze longitudinal clusters and identify causal mechanisms by which social, behavioral, and economic interventions can reduce survival differences.

## Introduction

Social risk factors for disease (such as material well-being, beliefs, or emotional states) are believed to influence one's health and longevity [1–5]. Two important social risk factors—income and educational attainment—have been shown to be linked to health or survival using quasi-experimental and/or randomized-controlled trials [6–12].

While most social risk factors cannot be experimentally tested, it is possible to improve on associational studies. Over the past 40 years, blacks and whites have differed with respect to material well-being, beliefs, emotional states, behaviors, and survival rates to varying degrees [13–18]. For example, while the 1990s saw health disparities between blacks and whites grow, the 2000s are notable for decreasing health disparities by race [19]. These variations by race over time can help us understand whether abstract health threats, such material well-being or emotional states, might concretely influence survival. Period differences in survival over time and by race are not as subject to endogeneity as prospective multivariate models.

For example, schooling has long been recognized as the primary determinant of one's access to material resources [20]. Local school taxes can fluctuate in minority neighborhoods according to macro-social trends (e.g., urbanization), economic trends (e.g., recessions or booms), or political priorities (e.g., the extent to which politicians invest in policies that produce social equity). Therefore changes in access to educational opportunities by race plausibly influence poverty rates and adult health (in the intermediate term) and survival (in the long term) [21].

While poverty rates and educational attainment can also differ between blacks and whites at different points in time, a lot less is known about other "social" risks for health and longevity. Nevertheless, they can differ by race over time. Changes social capital (e.g., trust in others) appear to be shaped by media events and sub-cultural norms [22]. Broader belief structures, like religion, can also fluctuate over time [23–25]. Likewise, changes in emotional or affective states or behaviors can be influenced by social trends, such as the time one spends at work rather than with friends and family [26].

One's belief structure or affective states can also plausibly produce broad impacts on health and longevity. For example, a collective sense of fairness can lead to funding for education or health services, while a distrust in institutions can limit one's interaction with the health system [27, 28]. A belief in God might influence both bridging social capital and one's core beliefs (e.g. trust in others or a sense of that others are fair) [29].

Educational attainment and social capital are also thought to be intertwined with emotional states, such as happiness [30–32]. Negative emotional states can increase stress [33, 34]. Stress, in turn, disrupts normal physiologic functions, such as the maintenance of blood pressure

[35]. Apart from directly altering human physiology, stress can also change behaviors, such as the maintenance of a healthy diet [36–38]. Negative emotional states are also plausibly related to other behaviors, such as sexual intimacy [39].

Over the past 4 decades, the United States has experienced historical shifts in mortality patterns alongside major shifts in material well-being (e.g., income), beliefs (e.g., in religion or God), emotional states (e.g., happiness), and behaviors (e.g., sex frequency) [19]. These variables are both patterned by race over time [13–18] and differ between generations. In this study, we explored whether racial differences in selected measures of material well-being, beliefs, emotional states, and behaviors also could partially explain racial differences in survival using the General Social Survey-National Death Index (GSS-NDI) 1978–2014.

## Methods

### Data

We used the 2014 GSS-NDI, which contains data from the GSS for different sociological variables from 1978 to 2010 merged with death certificate data, allowing for ascertainment of vital status through 2014 [40].The GSS-NDI is a nationally-representative sample of non-institutionalized US adults who were at least 18 years of age when surveyed.

The GSS-NDI contains surveys conducted annually from 1978 to 1994 (with an exception of 1979, 1981, and 1992), and biannually from 1994 until 2010. At the time of paper development, the GSS-NDI data were housed by the authors at Columbia University. The data are now housed at the National Opinion Research Center and require prior approval for use, as they have been reclassified as sensitive data [41].

The GSS interviewers conducted face-to-face interviews. Response rates ranged from 70% to 82%, depending on the survey year. The survey, conducted by the National Opinion Research Center (NORC) at the University of Chicago, utilizes a multi-stage probability sample.

We limited the sample to US native-born persons to avoid confounding by immigration status. This was done both because immigrants may hold different perceptions of our measures than native-born Americans of the same race, and because immigrants tend to live longer than native-born Americans [42]. For our analyses, we further limited our sample to white and black racial groups. Our GSS-NDI analyses did not make distinction between ethnic backgrounds within a given racial group.

### Selected measures of material well-being, beliefs, emotional states, and behaviors

Our select measures were self-rated health, happiness, trust in others, feeling that society is fair, belief in God, frequency of sexual intercourse, educational attainment, and poverty. These measures were collected over most years during the time span of the study (1978 to 2010).

Educational attainment was coded as 1 for people that claimed to have at least a high school degree, and 0 otherwise. Using the US Department of Health and Human Services Poverty Guidelines, the respondent's household income adjusted to the year 2000, and family size we created a variable to determine whether a respondent's household income fell above or below the poverty line. For self-rated health, in the GSS-NDI, the question takes the form "Would you say your own health, in general, is excellent, good, fair, or poor?" This variable was not asked in the GSS-NDI for the years 1978, 1983, and 1986. We coded the response variable as 0 for either 'poor' or 'fair health', and 1 for either 'good' or 'excellent health.' Happiness was measured using the question, "Taken all together, how would you say things are these days-

would you say that you are very happy, pretty happy, or not too happy?" We coded the response variable as 0 for 'not too happy', and 1 for either 'pretty' or 'very happy.'

Trust in others and the perception that one is being treated fairly are two beliefs that are plausibly linked to emotional states and are also key measures of social capital. Trust was measured by the question, "Generally speaking, would you say that most people can be trusted or that you can't be too careful in dealing with people?" The response variable was coded as 0 for 'cannot trust' to identify those who strictly believed that people couldn't be trusted. Otherwise, it was coded as 1 for those whose response was either 'depends' or 'can trust'. Likewise, the respondent's perception of societal fairness was captured as, "Do you think most people would try to take advantage of you if they got a chance, or would they try to be fair?" This variable was coded as 0 for those responding 'take advantage', and 1 for those responding either 'depends' or 'fair.' Such beliefs may be closely tied to religious beliefs. One's belief in God can increase social capital through religious services attendance, and religion may influence dietary and other health risks associated with premature mortality [29]. The respondent's belief in God was captured as, "Please look at this card and tell me which statement comes closest to expressing what you believe about God?" This variable was measured annually from 1987 onward (with an exception of 1989 and 2002). The variable was coded as 1 for those responding 'know God exists', and 0 otherwise.

Central to all of these variables is the concept of social capital. While we are able to measure some aspect of social capital directly (i.e., trust in others and the sense that people try to be fair), we do not have a strong measure of inter-personal intimacy or bonding. We therefore chose to include the frequency of sexual intercourse as a measure of well-being. This variable was captured by the question, "About how often did you have sex during the last 12 months?" This variable was measured from 1988 onward. The variable was coded as 1 for sexually active people, if their response was either 'weekly', or '2–3 per week', or '4+ per week'. Otherwise, it was coded as 0.

Each record in the GSS survey was linked to the respondents' vital status using a matching algorithm based on a set of parameters, including Social Security number, first and last name, birth date, race, and gender [43].

## Statistical framework

Our descriptive statistical analyses aimed to quantify the trends in racial inequalities in our measures of well-being and survival rates over time. It is comprised of two steps.

First, we conducted an APC analysis embedded within a multilevel modeling approach to quantify the trends in our select measures over time for blacks vs. whites. We used a generalized linear mixed-effects model with binomial likelihood and logit link to estimate APC trends for each measure. As our intention was to quantify the descriptive trends of such variables for blacks relative to whites, we only chose age, sex (male vs. female), and race (whites vs. blacks) as our independent covariates in a random-intercept, random-slope (race) model. The random effects for intercept and slope were estimated for different survey years and birth cohorts. In addition, we coded our birth cohorts in five-year intervals shown in the S1 Appendix (S1 Table in S1 Appendix) as suggested by Yang [44]. Our statistical model assumed the following form:

$$Wellbeing_{ijk} \sim binom(1, p_{ijk}),$$

$$\text{Log}\Big(odds(p_{ijk})\Big) = \beta_{0jk} + \beta_1.Age_{ijk} + \beta_2.Male_{ijk} + \beta_{3jk}.White_{ijk},$$

$$\beta_{0jk} = \beta_0 + \beta_{0j} + \beta_{0k},$$

$$\beta_{3jk} = \beta_3 + \beta_{3j} + \beta_{3k},$$

where $i$ (= 1,...,$I_{jk}$), $j$ (= 1,...,$J$), and $k$ (= 1,...,$K$) represents individuals, period, and cohort, respectively. $\beta_0$ is the expected value of a given well-being variable averaged over all years and cohorts given all other independent covariates set to zero. $\beta_{0j}$ and $\beta_{0k}$ allow for a random intercept that varies across periods, and cohorts, respectively, on a given well-being variable when all other independent covariates set to zero. Similarly, $\beta_{3j}$ and $\beta_{3k}$ allow for random slopes of the association between race and the variable of interest across periods and cohorts respectively. The random variance components of period and cohort for both the intercept and the slope (race) followed a multivariate normal distribution with a mean of zero.

Second, we ran a mixed-effects Cox proportional hazards model to quantify the trends in the hazard ratio (HR) of death for blacks vs. whites over time. Survival was estimated as number of years from birth to the age at death as recorded in the NDI, or, for those with no death record, to the participants' current age. Our model was restricted to age, sex, and race as independent predictors, in a random-intercept (baseline hazard), random-slope (race) frailty model to capture the potential variations in the survival of blacks vs. whites across different survey years and birth cohorts. Our mixed-effects Cox model took the form:

$$\lambda_{ijk}(t) = \lambda_0(t)e^{\beta_{0jk}+\beta_1.Age_{ijk}(t)+\beta_2.Male_{ijk}+\beta_{3jk}.White_{ijk}},$$

$$\beta_{0jk} = \beta_0 + \beta_{0j} + \beta_{0k},$$

$$\beta_{3jk} = \beta_3 + \beta_{3j} + \beta_{3k},$$

where $i$ (= 1,...,$I_{jk}$), $j$ (= 1,...,$J$), and $k$ (= 1,...,$K$) represents individuals, periods, and cohorts, respectively. $\lambda_0(t)$ is the unspecified baseline hazard function. $e^{\beta_0}$ is the baseline hazard component averaged across all periods and birth cohorts. $e^{\beta_{0j}}$ and $e^{\beta_{0k}}$ allow for a random baseline hazard of death associated with the $j$th period, and the $k$th cohort, respectively. $e^{\beta_3}$ is the HR of death for whites vs. blacks averaged across all periods and birth cohorts. $e^{\beta_{3j}}$ and $e^{\beta_{3k}}$ allow for a random HR of death for whites vs. blacks across periods and cohorts, respectively. The random variance components of periods and cohorts for both the intercept and the slope (race) were modeled with a multivariate normal distribution with a mean zero. We separately adjusted for variations in each of our select measures of well-being to capture their impacts on differences in survival of blacks and whites over time.

## Results

We detailed the summary statistics associated with each variable (including the mean and sample size available for analysis) for each survey year of interest in Table 1. We also reported the distribution and crude rates of the select measures across birth cohorts in the S1 Appendix (S1 Table in S1 Appendix).

### Period- and cohort-adjusted effects of age, sex, and race on measure of well-being

Table 2 shows the main results of the APC models adjusted for age, sex, period, and cohort effect. The variables amongst our select measures that differed by race at $p < 0.05$ were self-

**Table 1. The distribution and summary statistics of the select well-being measures of our study across different years in the GSS-NDI.**

| Birth Cohort | Age | Gender | Race | Income | Education | Sex frequency | Believing in God | Fair | Trust | Happy | Health |
|---|---|---|---|---|---|---|---|---|---|---|---|
| | | (male) | (white) | (above poverty line) | (high school or above) | (high sex frequency) | (know God exists) | (depends or fair) | (depends or can trust) | (pretty or very happy) | (good or excellent) |
| | Mean (SD) | N (%) | N (%) | N (%) | N (%) | N (%) | N (%) | N (%) | N (%) | N (%) | N (%) |
| **1978** | 43.94 (17.76) | 598 (0.42) | 1,267 (0.89) | 1132 (0.85) | 985 (0.69) | - | - | 976 (0.70) | 613 (0.43) | 1,266 (0.90) | - |
| **1980** | 44.81 (17.70) | 528 (0.44) | 1,084 (0.91) | 954 (0.86) | 831 (0.70) | - | - | 755 (0.64) | 572 (0.48) | 1,045 (0.88) | 879 (0.74) |
| **1982** | 44.64 (18.09) | 663 (0.41) | 1,155 (0.71) | 1158 (0.78) | 1,095 (0.68) | - | - | - | - | 1,389 (0.86) | 1,166 (0.72) |
| **1983** | 44.10 (17.35) | 547 (0.44) | 1,108 (0.88) | 959 (0.84) | 924 (0.74) | - | - | 797 (0.64) | 252 (0.40) | 1,080 (0.88) | - |
| **1984** | 44.06 (17.88) | 536 (0.41) | 1,143 (0.88) | 1008 (0.84) | 956 (0.74) | - | - | 843 (0.66) | 647 (0.50) | 1,119 (0.88) | 999 (0.77) |
| **1985** | 45.95 (17.98) | 616 (0.45) | 1,224 (0.90) | 1051 (0.84) | 1,005 (0.74) | - | - | - | - | 1,208 (0.89) | 1,039 (0.77) |
| **1986** | 45.58 (17.77) | 530 (0.43) | 1,087 (0.87) | 967 (0.85) | 917 (0.74) | - | - | 815 (0.66) | 495 (0.40) | 1,081 (0.88) | - |
| **1987** | 44.79 (17.63) | 689 (0.43) | 1,133 (0.71) | 1191 (0.8) | 1,199 (0.75) | - | - | 927 (0.59) | 674 (0.42) | 1,361 (0.87) | 1,222 (0.76) |
| **1988** | 45.55 (18.33) | 571 (0.43) | 1,156 (0.87) | 1015 (0.84) | 1,015 (0.76) | - | 839 (0.64) | 591 (0.67) | 390 (0.44) | 1,189 (0.91) | 656 (0.49) |
| **1989** | 45.87 (17.84) | 583 (0.43) | 1,206 (0.89) | 1056 (0.87) | 1,066 (0.79) | 547 (0.59) | - | 578 (0.64) | 407 (0.45) | 1,212 (0.91) | 702 (0.52) |
| **1990** | 46.61 (18.21) | 531 (0.44) | 1,072 (0.88) | 946 (0.87) | 976 (0.80) | 218 (0.56) | - | 500 (0.64) | 328 (0.42) | 1,105 (0.92) | 618 (0.51) |
| **1991** | 45.85 (17.87) | 573 (0.42) | 1,180 (0.87) | 1037 (0.84) | 1,094 (0.81) | 513 (0.59) | 749 (0.63) | 585 (0.64) | 394 (0.43) | 1,209 (0.90) | 665 (0.49) |
| **1993** | 46.59 (17.51) | 601 (0.43) | 1,232 (0.89) | 1070 (0.84) | 1,131 (0.82) | 554 (0.57) | 867 (0.67) | 577 (0.64) | 362 (0.40) | 1,238 (0.89) | 721 (0.52) |
| **1994** | 46.20 (17.10) | 1,140 (0.43) | 2,309 (0.87) | 2043 (0.87) | 2,223 (0.84) | 1,040 (0.56) | 764 (0.65) | 1,074 (0.60) | 674 (0.38) | 2,328 (0.88) | 1,384 (0.52) |
| **1996** | 45.30 (17.04) | 1,105 (0.44) | 2,145 (0.85) | 1943 (0.87) | 2,142 (0.85) | 1,051 (0.59) | - | 954 (0.58) | 652 (0.39) | 2,206 (0.88) | 1,689 (0.67) |
| **1998** | 46.36 (17.22) | 1,046 (0.44) | 2,013 (0.85) | 1838 (0.87) | 2,003 (0.85) | 850 (0.55) | 658 (0.63) | 986 (0.61) | 841 (0.43) | 2,072 (0.88) | 1,858 (0.78) |
| **2000** | 46.91 (17.63) | 1,023 (0.44) | 1,953 (0.84) | 1767 (0.87) | 1,967 (0.85) | 807 (0.56) | 631 (0.65) | 921 (0.61) | 653 (0.43) | 2,051 (0.90) | 1,478 (0.63) |
| **2002** | 46.95 (17.50) | 991 (0.44) | 1,909 (0.85) | 1750 (0.87) | 1,930 (0.86) | 745 (0.55) | - | 456 (0.61) | 317 (0.43) | 993 (0.88) | 1,157 (0.51) |
| **2004** | 46.91 (16.89) | 1,071 (0.45) | 2,038 (0.86) | 1843 (0.88) | 2,092 (0.88) | 766 (0.54) | - | 446 (0.60) | 317 (0.42) | 980 (0.87) | 898 (0.38) |
| **2006** | 48.57 (17.32) | 1,016 (0.43) | 2,005 (0.84) | 1797 (0.86) | 2,112 (0.89) | 481 (0.52) | 997 (0.63) | 642 (0.61) | 775 (0.38) | 2,085 (0.88) | 1,188 (0.50) |
| **2008** | 48.91 (17.40) | 753 (0.46) | 1,404 (0.85) | 1232 (0.85) | 1,453 (0.88) | 584 (0.56) | 1,004 (0.61) | 667 (0.60) | 413 (0.37) | 1,406 (0.86) | 807 (0.49) |
| **2010** | 48.97 (17.68) | 731 (0.43) | 1,412 (0.84) | 1203 (0.82) | 1,474 (0.88) | 529 (0.50) | 975 (0.59) | 659 (0.61) | 421 (0.38) | 1,435 (0.86) | 761 (0.45) |

GSS-NDI: General Social Survey-National Death Index.

**Table 2. Results of Age-Period-Cohort analyses indicating the odds ratio of the selected measures of well-being included in our analysis for whites versus blacks across different survey years and birth cohorts adjusted for age and sex.** Numbers in the bracket show the 95% Confidence Interval. General Social Survey-National Death Index (1978–2014).

| | Income (>poverty level) | Education (high school or above) | Sex frequency (high) | Belief in God (God exists) | Fair (depends or fair) | Trust (depends or can trust) | Happy (pretty or very happy) | Health (good or excellent) |
|---|---|---|---|---|---|---|---|---|
| **Intercept** | 0.183 (0.154, 0.216)*** | 0.466 (0.202, 1.077) | 1.252 (1.083, 1.448)*** | 1.786 (1.53, 2.085)*** | 1.637 (1.416, 1.893)*** | 0.708 (0.617, 0.812)*** | 5.145 (4.347, 6.089)*** | 8.15 (6.686, 9.934)*** |
| **Age** | 0.997 (0.995, 0.999)* | 1.02 (1.014, 1.026)*** | 1 (0.998, 1.002) | 0.999 (0.997, 1.001) | 1.001 (0.999, 1.003) | 1 (0.998, 1.002) | 0.996 (0.994, 0.998)** | 0.969 (0.966, 0.973)*** |
| **White** | 0.969 (0.856, 1.096) | 2.878 (2.347, 3.528)*** | 1.029 (0.928, 1.142) | 0.998 (0.877, 1.136) | 0.964 (0.864, 1.075) | 0.998 (0.908, 1.096) | 2.046 (1.773, 2.361)*** | 1.876 (1.632, 2.156)*** |
| **Male** | 0.966 (0.905, 1.03) | 0.966 (0.912, 1.022) | 1.015 (0.953, 1.081) | 1.034 (0.956, 1.118) | 1.01 (0.956, 1.067) | 1.017 (0.965, 1.072) | 1.048 (0.979, 1.123) | 1.055 (0.991, 1.124). |
| **Random effects in log scale (SD)** | | | | | | | | |
| **Period** | | | | | | | | |
| *Intercept* | 0.133 | 0.004 | 0.145 | 0.02442 | 0.231 | 0.185 | 0.119 | 0.053 |
| *White* | 0.063 | 0.103 | 0.089 | 0.10413 | 0.172 | 0.112 | 0.150 | 0.058 |
| **Cohort** | | | | | | | | |
| *Intercept* | 0.180 | 1.728 | 0.000 | 0.0299 | 0.000 | 0.012 | 0.170 | 0.215 |
| *White* | 0.153 | 0.400 | 0.000 | 0.00177 | 0.000 | 0.025 | 0.176 | 0.222 |
| **Number of observations** | 33976 | 37783 | 15604 | 11838 | 23603 | 24492 | 35148 | 25932 |
| **AIC** | 25572.8 | 31567.8 | 21913.9 | 14812.6 | 30434.5 | 32219.2 | 24115.9 | 25721.4 |

SD = standard deviation; AIC = Akaike Information Criterion

\***:p< = 0.001

\**: p< = 0.01; *: p< = 0.05;: P< = <0.1 (2-tailed tests).

rated health (odds ratio (OR) = 1.88 (95% confidence interval (CI): 1.63–2.16) for whites vs. blacks), happiness (OR = 2.05 (95% CI: 1.77–2.36) for whites vs. blacks), and high school graduation rate (OR = 2.88 (95% CI: 2.35–3.53) for whites vs. blacks). A one year increase in age was associated with self-rated health (OR = 0.97 (95% CI: 0.96–0.97)), happiness (OR = 0.99 (95% CI: 0.99–0.99)), high school graduation rate (OR = 1.02 (95% CI: 1.01–1.03)), and the odds of living above the poverty line (OR = 0.99 (95% CI: 0.99–0.99)). Our measures were not influenced by gender (Table 2).

## Period effects on the select measures of well-being

Fig 1 shows trends for blacks relative to whites for our selected measures across different survey years adjusted for age, sex, and birth cohort effects. In this figure, the ORs are calculated using rates for blacks as a reference group in the earliest available survey year (e.g., 1980 for self-rated health). Therefore, in Panel A, we see that the odds ratio of having good health in 1980 was about 70% higher for whites than blacks (OR = 1.7). Though the period effects for self-reported health were relatively stable between 1980 and 2000 for blacks, whites showed an upward trend through the 1980s. This trend then leveled off at around OR of 2.1 through the 1990s and early 2000s before a steep decline was observed between 2008 and 2010 (Fig 1-panel A). At its peak, the odds of self-rated health were 2.3 that of the reference group (blacks in 1980).

Similar to self-rated health, whites had the highest odds of happiness between 1988 and 1990. Relative to blacks in 1978, the odds of happiness among whites peaked at approximately 2.4. There appeared to be a slight gradual decline in happiness for whites relative to blacks in the 2000s (Fig 1-panel B).

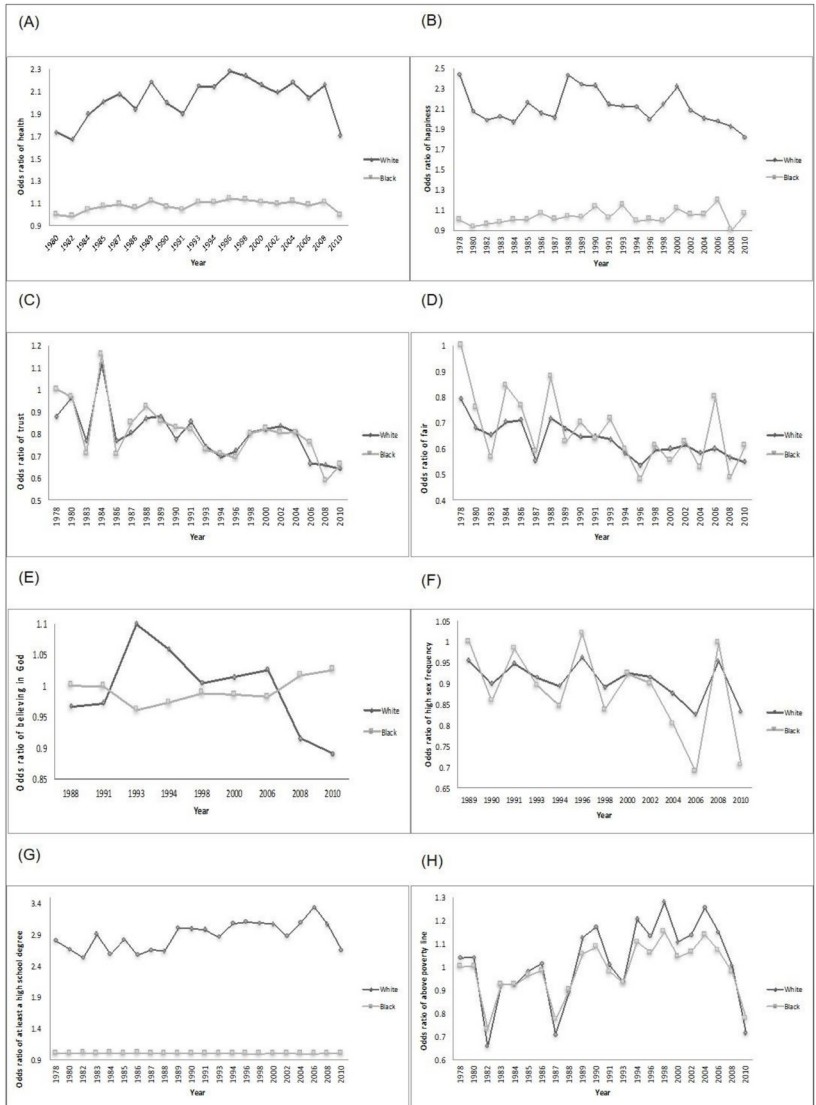

**Fig 1.** Period effects on odds ratio of well-being variables for whites vs. blacks (reference was the odds of well-being variables for blacks in the first year): (A) self-rated health; (B) happiness; (C) trust; (D) fair; (E) belief in God; (F) sex frequency; (G) High school degree; (H) Above poverty line. Analysis of data from the 2014 General Social Survey-National Death Index with survey data from 1978 to 2010 and mortality follow up to 2014.

For societal trust, feeling society is fair, and sex frequency, period effects appeared to have an overall downward trend for whites and blacks alike from 1980s to 2010 (Fig 1-panels C and D and F). Starting 1993, the period effects for belief in God appeared to have a downward trend for whites. By contrast, the odds of self-reporting a belief in God slightly increased for blacks over this period (Fig 1-panel E). Period effects on high school graduation rates had an overall increasing trend for whites relative to blacks (in reference year 1978) from 1978 (OR = 2.80) to 2006 (OR = 3.34), followed by a decreasing trend until 2010 (OR = 2.65; Fig 1-panel G). Finally, both whites and blacks showed an upward trend for living above the poverty line between 1980s and 2004, and a downward trend thereafter until 2010 (Fig 1-panel H).

## Cohort effects on the select measures of well-being

Cohort effects are presented in the S1 Appendix (S2 Fig in S1 Appendix). For all but self-rated health, happiness, and high school graduation rates, there were no noticeable cohort effects for blacks relative to whites. For self-rated health, there was a declining trend for birth cohorts between 1945 to 1960. In contrast, similar birth cohorts of blacks noticed an improving trend in that time frame. For happiness, while whites maintained a similar rate across different birth cohorts from 1899 to 1985, the relative happiness of blacks slightly declined. For high school graduation rates, both races experienced an increasing birth cohort effect from 1899 to 1970, followed by a decreasing effect thereafter.

## Survival effects of the select measures of well-being

After adjustment for age, period, cohort, sex, and race, amongst our select measures, self-rated health, happiness, trust, and high school graduation rates were statistically significantly associated with mortality. A self-rated health of "good" or "excellent" was associated with 29% reduction in hazard of death (HR = 0.71 [95% CI: 0.67–0.74]). Being "very happy" or "pretty happy" was associated with 16% reduction in hazard of death (HR = 0.84 [95% CI: 0.79–0.89]). In addition, the HR of death for high school graduation was 0.83 (95% CI: 0.79–0.87). For trust in others, the HR was 1.06 (95% CI: 1.01–1.11). Table 3 shows the results of our survival analyses.

**Table 3. Results of Age-Period-Cohort analyses (hazard ratio for mortality for whites versus blacks) across different survey years and birth cohorts adjusted for age, sex, and measures of well-being included in our analysis (95% Confidence Interval).** General Social Survey-National Death Index (1978–2014).

| | Mortality | Mortality adjusted for income | Mortality adjusted for education | Mortality adjusted for sex | Mortality adjusted for believing in God | Mortality adjusted for fair | Mortality adjusted for trust | Mortality adjusted for happiness | Mortality adjusted for self-rated health |
|---|---|---|---|---|---|---|---|---|---|
| **Age** | 1.051 (1.047, 1.055) *** | 1.051 (1.047, 1.055) *** | 1.05 (1.046, 1.054) *** | 1.062 (1.06, 1.064) *** | 1.062 (1.06, 1.064) *** | 1.053 (1.049, 1.058) *** | 1.053 (1.049, 1.058) *** | 1.049 (1.045, 1.053) *** | 1.048 (1.044, 1.052) *** |
| **White** | 0.733 (0.624, 0.86) *** | 0.718 (0.608, 0.848) *** | 0.762 (0.645, 0.9) *** | 0.6 (0.524, 0.687) *** | 0.662 (0.573, 0.763) *** | 0.736 (0.633, 0.856) *** | 0.731 (0.624, 0.857) *** | 0.728 (0.618, 0.856) *** | 0.748 (0.628, 0.891) *** |
| **Male** | 1.339 (1.29, 1.39) *** | 1.335 (1.284, 1.388) *** | 1.339 (1.29, 1.39) *** | 1.344 (1.258, 1.437) *** | 1.334 (1.233, 1.443) *** | 1.332 (1.271, 1.397) *** | 1.351 (1.289, 1.416) *** | 1.334 (1.282, 1.387) *** | 1.332 (1.274, 1.394) *** |
| **Adjusted variable of interest** | | 1.002 (0.947, 1.061) | 0.834 (0.799, 0.871) *** | 1.039 (0.972, 1.11) | 1.041 (0.96, 1.128) | 1.013 (0.967, 1.062) | 1.062 (1.013, 1.113) * | 0.838 (0.791, 0.887) *** | 0.707 (0.673, 0.742) *** |
| **Random effects in log scale (SD)** | | | | | | | | | |
| **Period** | | | | | | | | | |
| *Intercept* | 0.090 | 0.109 | 0.088 | 0.132 | 0.150 | 0.072 | 0.076 | 0.100 | 0.089 |
| *White* | 0.018 | 0.072 | 0.059 | 0.089 | 0.074 | 0.012 | 0.006 | 0.071 | 0.082 |
| **Cohort** | | | | | | | | | |
| *Intercept* | 0.366 | 0.346 | 0.382 | 0.052 | 0.095 | 0.294 | 0.301 | 0.338 | 0.364 |
| *White* | 0.311 | 0.313 | 0.321 | 0.168 | 0.154 | 0.270 | 0.292 | 0.306 | 0.322 |
| **Number of observations (number of events)** | 37854 (11361) | 33976 (10302) | 37783 (11325) | 15604 (3469) | 11838 (2581) | 23603 (7443) | 24492 (7414) | 35148 (10835) | 25932 (7700) |
| **AIC** | 9848.35 | 8834.92 | 9877.71 | 4105.54 | 2770.42 | 6316.15 | 6370.1 | 9405.25 | 6911.86 |

SD: standard deviation; AIC: Akaike Information Criterion.

***: p< = 0.001

**: p< = 0.01

*: p< = 0.05˙: P< = <0.1 (2-tailed tests).

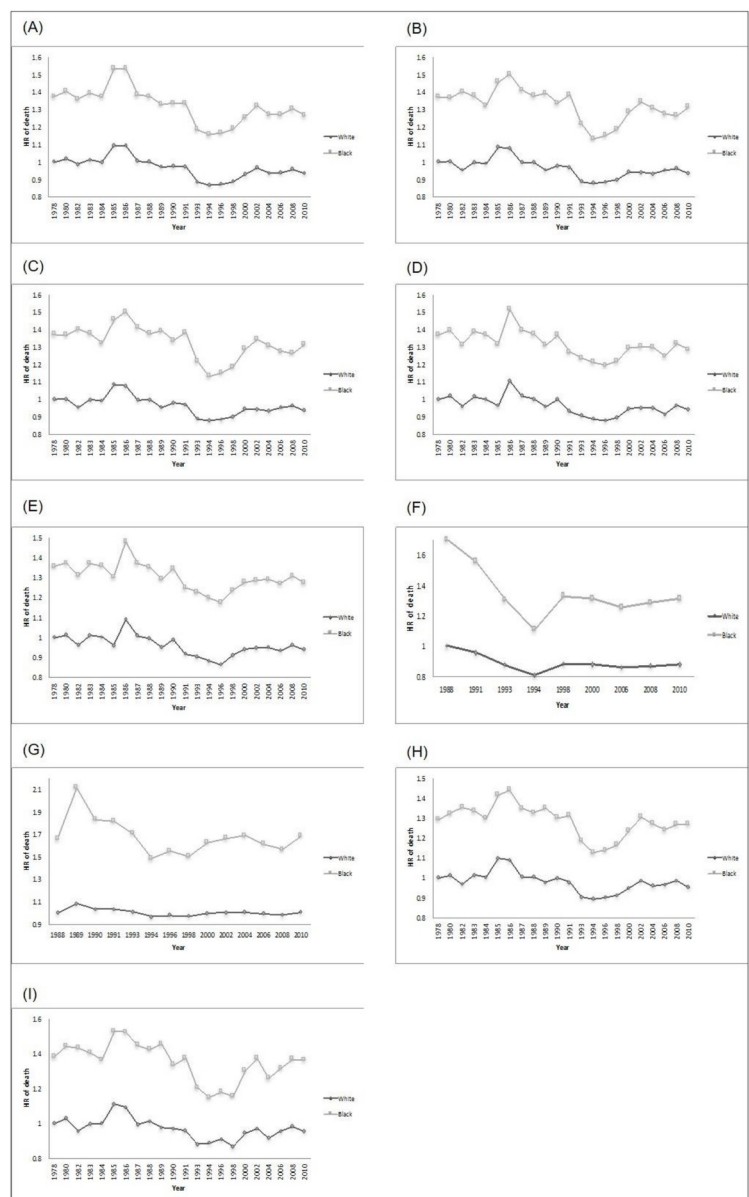

**Fig 2.** Period effects on hazard ratio of mortality for blacks vs. whites (reference was the hazard of mortality for whites in the first year): (A) Unadjusted mortality; and adjusted for (B) self-rated health; (C) happiness; (D) trust; (E) fair; (F) belief in God; (G) sex frequency; (H) High school degree; (I) Above poverty line. Analysis of data from the 2014 General Social Survey-National Death Index with survey data from 1978 to 2010 and mortality follow up to 2014.

Fig 2 shows the trends in racial survival differences without (panel A) and with adjustment for variations in our select measures of well-being (panels B to H). In this figure, the HR of death for blacks and whites across different survey years is adjusted for age, sex, and birth cohort effects. Whites in 1978 served as the reference group. Whites consistently had higher rates of survival relative to blacks across all years. Relative differences in black/white survival rates followed similar patterns to those observed in racial differences in self-rated health, happiness, and educational attainment across survey years. That is, generally those survey years with greater differences in self-rated health, happiness, and educational attainment between blacks and whites were also associated with larger differences in survival. The highest HR of

death between blacks and whites that was observed in the year 1989 (HR = 1.57). This difference in survival by race would have dropped to 1.47, 1.53, and 1.49 respectively had there not been racial differences in self-rated health, happiness, and high school graduation rates. The cohort effects on survival trends of blacks vs. whites are presented in the S1 Appendix (S3 Fig in S1 Appendix). Trends by cohort were generally similar to trends by period.

## Discussion

It is generally very difficult to move beyond associational studies for measures of material well-being, beliefs, emotional states, and behaviors [6]. Our study attempts to do so by accounting for age, period, and cohort effects on survival differences between blacks and whites in a unified model that spans decades of survey data. When conducting trend analyses such as ours, it is important to remember that a person who was 50-years-old in 1980 would likely have a very different set of beliefs from a 50-year-old today. For this reason, we deploy an age-period-cohort model. For our survival analysis, we used a mixed-effects Cox proportional hazard model, which provides a more solid understanding of survival differences between blacks and whites by considering the effect of time to event as opposed to simple survival rate ratios.

We find that, with respect to self-rated health, happiness, and educational attainment, whites were consistently more socially advantaged than blacks over our study period. Nevertheless, differences by race changed both by cohort and period. For example, although whites consistently self-rate as happier than blacks, whites have shown relative declines in happiness across since 1978. Likewise, self-rated health began to decline for whites relative to blacks in the 1990s. While differences in survival between blacks and whites remained large throughout our period of analysis, these selected measures of well-being that we were able to include in our analysis did explain some of the black/white differences in survival. This was true even after adjusting for self-rated health, happiness, and educational attainment.

On the other hand, we found that black/white differences in perceptions of trust in others, perceptions that people try to be fair, belief in God, or frequency of sexual intercourse did not substantially vary over the study period. Therefore, we would not expect to see fluctuations in survival associated with these measures. Accordingly, with the exception of the analysis on trust in others, we did not observe statistically-significant changes in survival associated with these variables. Associations between trust and health have been previously observed, and our models may have simply been sensitive to small fluctuations or may have been confounded by third variables over time [45]. Future waves of the GSS-NDI may allow for a deeper exploration of our well-being measures, as changes in self-reported health and happiness materialized toward the end of our analysis.

Our findings are in line with those of previous studies that looked at the trends in well-being. There is already evidence that the rapid changes in societal attitudes towards same-sex marriage in the US produced positive effects on the mental and physical health of same-sex couples since the early 1990s [46–48], and that changes in racial prejudice can influence the survival of both blacks and whites [49]. Other studies show similar age-standardized probabilities of poor/fair health, as well as period effects for happiness, trust, and sexual frequency in the US [50, 51].

The most important limitation of our study is that we could only choose measures of material well-being, beliefs, emotional states, and behaviors for which we had consistent data over time. It could be that there are other variables that capture important dimensions of racial differences in well-being that our included variables do not. Also, our study was based on a survey, which necessarily includes selection bias and does not capture institutionalized adults. This becomes important when considering that blacks are much more likely to have contact

with the criminal justice system than whites are, and that rates of contact vary over time [52]. In addition, our study was confined by inherent limitations of APC analyses through multi-level modeling [44].

Our study sought to analyze the trends of self-rated health, happiness, trust in others, feeling that society is fair, belief in God, frequency of sexual intercourse, educational attainment, and poverty for whites relative to blacks and their association with racial differences in mortality. Future steps are needed to analyze longitudinal clusters and identify causal mechanisms by which social, behavioral, and economic interventions can reduce racial survival disparities.

## Supporting information

**S1 Appendix.**
(DOCX)

## Author Contributions

**Conceptualization:** Zafar Zafari, Katherine M. Keyes, Sharifa Z. Williams, Peter Alexander Muennig.

**Formal analysis:** Zafar Zafari.

**Funding acquisition:** Zafar Zafari, Peter Alexander Muennig.

**Methodology:** Zafar Zafari, Katherine M. Keyes, Sharifa Z. Williams, Peter Alexander Muennig.

**Resources:** Peter Alexander Muennig.

**Software:** Zafar Zafari.

**Supervision:** Peter Alexander Muennig.

**Validation:** Zafar Zafari.

**Visualization:** Zafar Zafari, Boshen Jiao.

**Writing – original draft:** Zafar Zafari.

**Writing – review & editing:** Katherine M. Keyes, Boshen Jiao, Sharifa Z. Williams, Peter Alexander Muennig.

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
