## [Decision Letter · Decision Letter 0]

1 May 2020

PONE-D-20-01212

Differences between blacks and whites in well-being, beliefs, emotional states, behaviors and survival, 1978-2014

PLOS ONE

Dear Dr Zafari,

Thank you for submitting your manuscript to PLOS ONE. After careful consideration, we feel that it has merit but does not fully meet PLOS ONE’s publication criteria as it currently stands. Therefore, we invite you to submit a revised version of the manuscript that addresses the points raised during the review process.

We would appreciate receiving your revised manuscript by Jun 15 2020 11:59PM. To enhance the reproducibility of your results, we recommend that if applicable you deposit your laboratory protocols in protocols.io, where a protocol can be assigned its own identifier (DOI) such that it can be cited independently in the future. For instructions see: http://journals.plos.org/plosone/s/submission-guidelines#loc-laboratory-protocols

We look forward to receiving your revised manuscript.

Kind regards,

Sabine Rohrmann

Academic Editor

PLOS ONE

Journal Requirements:

2. In your Methods section, please provide additional information on how survival was calculated.

5. Please include captions for your Supporting Information files at the end of your manuscript, and update any in-text citations to match accordingly. Please see our Supporting Information guidelines for more information: http://journals.plos.org/plosone/s/supporting-information

Reviewers' comments:

Reviewer's Responses to Questions

**Comments to the Author**

1. Is the manuscript technically sound, and do the data support the conclusions?

Reviewer #1: Yes

2. Has the statistical analysis been performed appropriately and rigorously? 

Reviewer #1: Yes

3. Have the authors made all data underlying the findings in their manuscript fully available?

Reviewer #1: Yes

4. Is the manuscript presented in an intelligible fashion and written in standard English?

Reviewer #1: Yes

5. Review Comments to the Author

Reviewer #1: This is an interesting paper which examines black white disparities in metrics of well-being and mortality over time. The authors note that whites have more favorable metrics of well-being compared to blacks and well as lower rates of mortality. After accounting for differences in well-being the black-white disparity in survival is slightly attenuated suggesting that intervening on disparities in well-being may reduce survival disparities between whites and blacks. There are a few points that are worth addressing:

Intro:

“the blacks and whites” sounds strange, suggest “blacks and whites”

Methods:

Is the GSS-NDI a random digit dial survey? Is it a mailed survey? Is it a weighted complex/survey design? Is there any info on the proportion surveyed who accept?

You didn’t describe how death was assessed, you make clear that it is NDI but it warrants a separate mention and at least a reference under variables.

Statistical framework:

You mention you limit the sample size to exclude immigrants. What is your analytical sample size? Please explicitly make mention of your inclusion/exclusion criteria.

For example, you focus on black white disparities, are there non black or white participants in this survey? Did you exclude Hispanics? Please expand.

Results:

The last line on page 9 is unclear, please clarify what the happiness, HS graduation, and poverty lines ORs are associated with? Is one year of age the predictor? It is not clear from how the sentence is written.

Tables and Figures:

Again we have no sample sizes, how do we know how large this survey is?

I understand that it is difficult to show data for many years, but there is no indication of the distribution of the variables of interest (i.e. how common is poverty or what is the proportion of the sample that is black or white etc)? These stats would be typical of a Table 1 in most papers

Table 1 and 2: why show the log odds? Why not make interpretation easier and show Odds ratios and hazards ratios?

The figures are illegible in the version sent to reviewers.

6. PLOS authors have the option to publish the peer review history of their article (what does this mean?). If published, this will include your full peer review and any attached files.

Reviewer #1: No

---

## [Author Response · Author response to Decision Letter 0]

21 Jul 2020

Comments in response to editorial requests

[Response] There have been no changes in the financial disclosure.

To enhance the reproducibility of your results, we recommend that if applicable you deposit your laboratory protocols in protocols.io, where a protocol can be assigned its own identifier (DOI) such that it can be cited independently in the future. For instructions see: http://journals.plos.org/plosone/s/submission-guidelines#loc-laboratory-protocols

 A rebuttal letter that responds to each point raised by the academic editor and reviewer(s). This letter should be uploaded as separate file and labeled 'Response to Reviewers'.

 [Response] We have provided a separate point-by-point response letter addressing all the comments by the editorial office and the reviewers.

 A marked-up copy of your manuscript that highlights changes made to the original version. This file should be uploaded as separate file and labeled 'Revised Manuscript with Track Changes'.

 [Response] We have uploaded a ‘Revised Manuscript with Track Changes’ as suggested.

 An unmarked version of your revised paper without tracked changes. This file should be uploaded as separate file and labeled 'Manuscript'.

 [Response] We have uploaded the ‘Manuscript’ file as suggested.

We look forward to receiving your revised manuscript.

Kind regards,

Sabine Rohrmann

Academic Editor

PLOS ONE

Journal Requirements:

[Response] We have ensured the compliance of our paper with PLOS ONE’s style requirements.

2. In your Methods section, please provide additional information on how survival was calculated.

 [Response] We have now added details on how death was assessed in NDI and the details of our cox proportional hazard models to the Methods section. Please see the extensive changes under “Statistical Framework.” We have added the following:

“Each record in the GSS survey was linked to the respondents’ vital status using a matching algorithm based on a set of parameters, including Social Security number, first and last name, birth date, race, and gender.”

We also added:

“Second, we ran a mixed-effects Cox proportional hazards model to quantify the trends in the hazard ratio (HR) of death for blacks vs. whites over time. Survival was estimated as number of years from birth to the age at death as recorded in the NDI, or, for those with no death record, to the participants’ current age. Our model was restricted to age, sex, and race as independent predictors, in a random-intercept (baseline hazard), random-slope (race) frailty model to capture the potential variations in the survival of blacks vs. whites across different survey years and birth cohorts. Our mixed-effects Cox model took the form:

λ_ijk (t)=λ_0 (t) e^(β_0jk+β_1.Age_ijk (t)+β_2.Male_ijk+β_3jk.White_ijk ), 

β_0jk=β_0+β_0j+β_0k, 

β_3jk=β_3+β_3j+β_3k, 

where i (=1,…,I_jk), j (=1,…,J), and k (=1,…,K) represents individuals, periods, and cohorts, respectively. λ_0 (t) is the unspecified baseline hazard function. e^(β_0 ) is the baseline hazard component averaged across all periods and birth cohorts. e^(β_0j ) and e^(β_0k ) allow for a random baseline hazard of death associated with the jth period, and the kth cohort, respectively. e^(β_3 ) is the HR of death for whites vs. blacks averaged across all periods and birth cohorts. e^(β_3j ) and e^(β_3k ) allow for a random HR of death for whites vs. blacks across periods and cohorts, respectively. The random variance components of periods and cohorts for both the intercept and the slope (race) were modeled with a multivariate normal distribution with a mean zero. We separately adjusted for variations in each of our select measures of well-being to capture their impacts on differences in survival of blacks and whites over time. ”

[Response] At the time that the manuscript was under preparation, we were in the process of transferring the identified data to the National Opinion Research Center (NORC). Because the data contain vital status, NORC has determined that they are identifiable data and has placed restrictions upon their use. We have amended this statement in the paper, and now refer the reader to the proper contacts within NORC to apply for access to the data. Please see “Methods” second paragraph. 

 If there are ethical or legal restrictions on sharing a de-identified data set, please explain them in detail (e.g., data contain potentially identifying or sensitive patient information) and who has imposed them (e.g., an ethics committee). Please also provide contact information for a data access committee, ethics committee, or other institutional body to which data requests may be sent.

[Response] In our revised cover letter, we now direct the editor and reviewer to the sensitive data file policy for the GSS-NDI at NORC, which can be found at: https://gss.norc.org/Documents/other/ObtainingGSSSensitiveDataFiles.pdf. 

[Response] As per above, the data are now classified as sensitive.

[Response] The corresponding author has now created an ORCID iD account as requested.

5. Please include captions for your Supporting Information files at the end of your manuscript, and update any in-text citations to match accordingly. Please see our Supporting Information guidelines for more information: http://journals.plos.org/plosone/s/supporting-information

[Response] The captions for the Supporting Information are now added at the end of the paper and in-text citations were updated accordingly.

Reviewers' comments:

Reviewer's Responses to Questions

Comments to the Author

1. Is the manuscript technically sound, and do the data support the conclusions?

Reviewer #1: Yes

2. Has the statistical analysis been performed appropriately and rigorously? 

Reviewer #1: Yes

3. Have the authors made all data underlying the findings in their manuscript fully available?

Reviewer #1: Yes

4. Is the manuscript presented in an intelligible fashion and written in standard English?

Reviewer #1: Yes

5. Review Comments to the Author

Reviewer #1: This is an interesting paper which examines black white disparities in metrics of well-being and mortality over time. The authors note that whites have more favorable metrics of well-being compared to blacks and well as lower rates of mortality. After accounting for differences in well-being the black-white disparity in survival is slightly attenuated suggesting that intervening on disparities in well-being may reduce survival disparities between whites and blacks. There are a few points that are worth addressing:

[Response] Thanks much for your comments. We have gone in-depth addressing all your comments and provided a detailed point-by-point response letter found below. 

1. Intro:

“the blacks and whites” sounds strange, suggest “blacks and whites”

[Response] Thank you. We have made corrections throughout.

2. Is the GSS-NDI a random digit dial survey? Is it a mailed survey? Is it a weighted complex/survey design? Is there any info on the proportion surveyed who accept?

[Response] Thank you for noting this oversight. In the revised version of the paper, we have now included this information to the Methods section. Please see the 3rd paragraph following the “Data” sub header.

3. You didn’t describe how death was assessed, you make clear that it is NDI but it warrants a separate mention and at least a reference under variables.

[Response] Thank you for the opportunity to clarify. Please see the paragraph immediately preceding the “Statistical Framework” sub header in the methods section.

4. Statistical framework:

You mention you limit the sample size to exclude immigrants. What is your analytical sample size? Please explicitly make mention of your inclusion/exclusion criteria.

For example, you focus on black white disparities, are there non black or white participants in this survey? Did you exclude Hispanics? Please expand.

[Response] Thank you for noting this. For the sample size, we now mention the summary statistics and sample size information in a newly created Table 1 and review this information in the results section. Please see a copy of the table in the paper and our response to your comment on “Tables&Figures” below. 

In addition, in the 4th paragraph of the Methods section (under the “Data” sub header), we provide additional justification for excluding foreign-born participants.

5. Results:

The last line on page 9 is unclear, please clarify what the happiness, HS graduation, and poverty lines ORs are associated with? Is one year of age the predictor? It is not clear from how the sentence is written.

[Response] Yes, we used a one-year increase in age to derive the ORs associated with happiness, HS graduation rate, and odds of living above poverty line. We have now made this clear and revised the corresponding section of the paper. 

6. Tables and Figures:

Again we have no sample sizes, how do we know how large this survey is?

I understand that it is difficult to show data for many years, but there is no indication of the distribution of the variables of interest (i.e. how common is poverty or what is the proportion of the sample that is black or white etc)? These stats would be typical of a Table 1 in most papers

[Response] Many thanks for the comment. We have added all the requested data and have added a new Table 1 describing sample sizes. Please note that the revised version of the study has 3 tables instead of the original 2 tables as per reviewer’s request. 

For cohort effects: please also note that we reported the distribution, and summary statistics, of the select wellbeing measures of our model in the Online Appendix Table S1 due to the lack of space in the paper body.

7. Table 1 and 2: why show the log odds? Why not make interpretation easier and show Odds ratios and hazards ratios?

The figures are illegible in the version sent to reviewers.

[Response] As recommended, we have now converted all those numbers and reported them in odds ratios, or hazard ratios, as opposed to log odds or log hazard ratios. Please see the updated Tables 2 and 3 in the revised manuscript. We have also converted the figures to high resolution images and will double check them for clarity in the Editorial Manager system.

---

## [Decision Letter · Decision Letter 1]

27 Aug 2020

Differences between blacks and whites in well-being, beliefs, emotional states, behaviors and survival, 1978-2014

PONE-D-20-01212R1

Dear Dr. Zafari,

We’re pleased to inform you that your manuscript has been judged scientifically suitable for publication and will be formally accepted for publication once it meets all outstanding technical requirements.

Kind regards,

Sabine Rohrmann

Academic Editor

PLOS ONE

Additional Editor Comments (optional):

Reviewers' comments:

Reviewer's Responses to Questions

**Comments to the Author**

1. If the authors have adequately addressed your comments raised in a previous round of review and you feel that this manuscript is now acceptable for publication, you may indicate that here to bypass the “Comments to the Author” section, enter your conflict of interest statement in the “Confidential to Editor” section, and submit your "Accept" recommendation.

Reviewer #1: All comments have been addressed

2. Is the manuscript technically sound, and do the data support the conclusions?

Reviewer #1: Yes

3. Has the statistical analysis been performed appropriately and rigorously? 

Reviewer #1: Yes

4. Have the authors made all data underlying the findings in their manuscript fully available?

Reviewer #1: Yes

5. Is the manuscript presented in an intelligible fashion and written in standard English?

Reviewer #1: Yes

6. Review Comments to the Author

Reviewer #1: This is an interesting paper which examines black white disparities in

metrics of well-being and mortality over time. The authors note that whites have more

favorable metrics of well-being compared to blacks and well as lower rates of mortality.

After accounting for differences in well-being the black-white disparity in survival is

slightly attenuated suggesting that intervening on disparities in well-being may reduce

survival disparities between whites and blacks. Thank you for your hard work addressing reviewer concerns.

7. PLOS authors have the option to publish the peer review history of their article (what does this mean?). If published, this will include your full peer review and any attached files.

Reviewer #1: No

---

## [Editor Report · Acceptance letter]

3 Sep 2020

PONE-D-20-01212R1 

Differences between blacks and whites in well-being, beliefs, emotional states, behaviors and survival, 1978-2014 

Dear Dr. Zafari:

I'm pleased to inform you that your manuscript has been deemed suitable for publication in PLOS ONE. Congratulations! Your manuscript is now with our production department. 

Kind regards, 

on behalf of

Dr. Sabine Rohrmann 

Academic Editor

PLOS ONE